# Healthcare utilisation in patients with long-term conditions during the COVID-19 pandemic: a population-based observational study of all patients across Greater Manchester, UK

Camilla Sammut-Powell ,[1,2] Richard Williams,[1,2,3] Matthew Sperrin ,[1] Owain Thomas,[4] N Peek,[1,2,3,5] Stuart W Grant[6]

For numbered affiliations see end of article.

**Correspondence to**
Dr N Peek;
niels.peek@manchester.ac.uk

## ABSTRACT

**Objectives** Data on population healthcare utilisation (HCU) across both primary and secondary care during the COVID-19 pandemic are lacking. We describe primary and secondary HCU stratified by long-term conditions (LTCs) and deprivation, during the first 19 months of COVID-19 pandemic across a large urban area in the UK.

**Design** A retrospective, observational study.

**Setting** All primary and secondary care organisations that contributed to the Greater Manchester Care Record throughout 30 December 2019 to 1 August 2021.

**Participants** 3 225 169 patients who were registered with or attended a National Health Service primary or secondary care service during the study period.

**Primary outcomes** Primary care HCU (incident prescribing and recording of healthcare information) and secondary care HCU (planned and unplanned admissions) were assessed.

**Results** The first national lockdown was associated with reductions in all primary HCU measures, ranging from 24.7% (24.0% to 25.5%) for incident prescribing to 84.9% (84.2% to 85.5%) for cholesterol monitoring. Secondary HCU also dropped significantly for planned (47.4% (42.9% to 51.5%)) and unplanned admissions (35.3% (28.3% to 41.6%)). Only secondary care had significant reductions in HCU during the second national lockdown. Primary HCU measures had not recovered to prepandemic levels by the end of the study. The secondary admission rate ratio between multi-morbid patients and those without LTCs increased during the first lockdown by a factor of 2.40 (2.05 to 2.82; p<0.001) for planned admissions and 1.25 (1.07 to 1.47; p=0.006) for unplanned admissions. No significant changes in this ratio were observed in primary HCU.

**Conclusion** Major changes in primary and secondary HCU were observed during the COVID-19 pandemic. Secondary HCU reduced more in those without LTCs and the ratio of utilisation between patients from the most and least deprived areas increased for the majority of HCU measures. Overall primary and secondary care HCU for some LTC groups had not returned to prepandemic levels by the end of the study.

## STRENGTHS AND LIMITATIONS OF THIS STUDY

⇒ This study includes data on over 3 million individuals, representing all patients registered with a general practitioner across an entire geographical area.
⇒ Both primary and secondary care services were analysed in this study.
⇒ Five surrogate markers of healthcare utilisation were considered in the primary care analyses.
⇒ Historical data prior to the start of the COVID-19 pandemic were lacking which limited the trend analyses that could be performed.
⇒ Data from secondary care providers were limited to only a subset of the population.

## INTRODUCTION

On 30 January 2020, the WHO declared a public health emergency of international concern with governments urged to prepare for global spread of COVID-19.[1] With case numbers increasing and the virus spreading globally, COVID-19 was characterised as a pandemic 6 weeks later and rapidly developed into a global public health emergency. As of 17 December 2021, approximately 273 million cases and 5.3 million COVID-19 associated deaths had been reported globally.[2] Governments across the world enacted a range of measures aimed at controlling the spread of the virus,[3] and increasing healthcare capacity.[4 5] Despite these measures, healthcare systems was overwhelmed and diversion of healthcare resources to address increased demand specific to COVID-19 has been required.[6 7] The impact of this diversion of resources on the care of patients with non COVID-19 illnesses was exacerbated by reduced staff availability due to COVID-19 infection among healthcare workers.[8]

Numerous studies have been undertaken to assess the impact of the pandemic on healthcare provision in a variety of settings. An analysis of UK general practitioner (GP) data demonstrated that diagnoses of common physical and mental health conditions decreased substantially early in the pandemic.[9] The number of urgent GP referrals for cancer fell by 60% in April 2020 compared with the same month in 2019.[10] Hospital administrative data have demonstrated a decline in patients presenting with acute coronary syndrome from mid-February 2020 onwards,[11] and a separate analysis demonstrated a 43% reduction in patients undergoing percutaneous coronary interventions for ST-elevation myocardial infarctions compared with previous years.[12] Modelling studies have suggested that approximately 28 000 000 elective surgical procedures were cancelled over a 12-week period of peak disruption caused by the pandemic.[13]

Most studies to date investigating the impact of the pandemic on healthcare utilisation (HCU) have assessed specific patient groups, largely focused on secondary care.[14] Changes to HCU during the COVID-19 pandemic for both primary and secondary care stratified across the range of long-term medical conditions (LTCs) and different levels of social deprivation have not previously been described. The Greater Manchester Care Record (GMCR) includes electronic health records from all primary and secondary care National Health Service (NHS) providers in the metropolitan county of Greater Manchester (GM). GM has been significantly affected by COVID-19,[15] and the GMCR provides a unique opportunity to study the impact of the pandemic on primary and secondary HCU in patients with LTCs across this defined urban area.

## METHODS
### Design and data source
This was a retrospective, observational, service evaluation using routinely collected data. The data analysed were from two sources: (1) HCU data from the GMCR which is an integrated patient record containing data from primary and secondary NHS services across GM and (2) contextual Government COVID-19 data[16] regarding the number of new COVID-19 cases and COVID-19-related hospital admissions.

### GM care record
The GMCR is populated with data from primary care (GPs), secondary care (acute and community hospitals), mental health trusts and social care organisations across an entire geographical region. A total of 9 secondary care organisations (including 12 hospitals), 3 mental health trusts and 10 clinical commissioning groups (CCGs) contribute data. The primary purpose of the GMCR is for direct patient care as it provides clinicians with information from other healthcare providers relevant to their patient encounters that would ordinarily be inaccessible.

However, it has also been made available in deidentified format for research relating to COVID-19.

### UK government COVID-19 data
Data regarding the number of new COVID-19 cases and COVID-19-related hospital admissions were collected by the UK government throughout the pandemic. The number of new cases by specimen date was extracted for Manchester and the number of COVID-19 admissions were extracted for each of the secondary acute providers serving the people with a Manchester CCG (MCCG), included within the GMCR. The data are freely available from https://coronavirus.data.gov.uk/details/download and full details of the data extraction are provided in online supplemental table S1.

### Data processing and approvals
All identifiable data including free text are redacted. Some non-identifying demographic data are available such as recorded gender, year of birth, lower layer super output area (LSOA), Index of Multiple Deprivation (IMD) and ethnicity. The University of Manchester is permitted to perform research on this data via a GM wide data protection impact assessment (DPIA). The basis for this DPIA is the control of patient information notice issued by the Secretary of State for Health and Social Care in March 2020 which allowed confidential patient information to be shared for the purposes of research into COVID-19.[17]

### Study populations and key time points
The main study population consisted of all patients that were registered with a GP within GM on 1 January 2020, herein defined as the GM population. The 1 January 2020 is the index study date. For the primary care analyses, the entire GM population were considered. However, secondary care data were only available for patients registered to a MCCG, hence, the secondary HCU analyses were limited to these people. The dates of the national lockdowns initiated in response to the COVID-19 pandemic were indicated in addition to Christmas week due to expected changes in HCU during these periods. The first national lockdown ran from 23 March 2020 to 11 May 2020, the second national lockdown ran from 5 November 2020 to 1 December 2020 and the third national lockdown ran from 6 January 2021 to the 8 March 2021.

### Long-term medical conditions
LTCs were defined as per Barnett et al[18] and were grouped into the following categories: cancer, cardiovascular, endocrine, gastrointestinal, musculoskeletal or skin, neurological, psychiatric, renal or urological, respiratory, sensory impairment or learning disability, and substance abuse. A resident was identified as being diagnosed with an LTC by interrogating the GMCR record prior to the index date. If an LTC was diagnosed after 1 January 2020, the patient was not recorded as having the LTC for this analysis. People who were identified as belonging to multiple LTC groups were assigned to each corresponding LTC group

and defined as multimorbid. The full list of LTCs and groupings are provided in online supplemental table S2.

### Index of Multiple Deprivation

The 2019 IMD is the official measure of relative deprivation provided by the Office for National Statistics which combines information from seven different domains to produce an overall relative measure of deprivation for each LSOA. Each LSOA is ranked from least to most deprived, and deciles of relative deprivation are generated.[19] For this study, the available IMD deciles were categorised into four groups, representing the most deprived (deciles 1–2), highly deprived (deciles 3–4), moderately deprived (deciles 5–6) and the least deprived LSOAs (deciles 7–10). The least deprived LSOA group consisted of four deciles to avoid multiple small groups because of the skew towards more deprived LSOA deciles within GM.

### Measuring HCU

For primary HCU, appointment data were not available within the GMCR. Surrogate markers of HCU were, therefore, evaluated consisting of first prescriptions and recording of healthcare information in the GP record. First prescriptions were identified by the issuing of any new prescription (non-repeat prescription) for an individual patient by a primary care healthcare professional and are herein referred to as new prescriptions. This measure was selected as the issuing of an incident prescription requires contact with a healthcare professional. Healthcare information recorded included: recoding of smoking status, measurements of cholesterol, blood pressure (BP), blood glucose (haemoglobin A1c (HbA1c)) and body mass index (BMI). The values of these measurements were not used in the analysis.

For secondary HCU, the number of acute provider admissions was evaluated. To enable population admission rates to be evaluated, a denominator was calculated by assigning each resident within MCCG to a secondary care provider according to the most common provider observed within their LSOA. In cases where the most common secondary provider was unclear one of the two most common providers was randomly assigned to that LSOA. Secondary care admissions were categorised into planned, unplanned, maternity, transfers and 'other' admissions, defined according to the admission type field available in the provider data. Daily aggregate level data counts of all utilisation measures were provided. A full description of the data processing applied is available at https://github.com/rw251/gm-idcr/tree/master/projects/001-Grant.

### Statistical analysis

Weekly totals of HCU data were evaluated for the entire population. The rate ratios (RRs) of utilisation in the weeks before and after the initiation of the first and second national lockdowns (first national lockdown: w/c 23 March 2020 vs w/c 9 March 2020; second national lockdown: w/c 9 November 2020 vs w/c 19 October 2020; w/c = week commencing) were estimated across all measures of HCU to determine the association between the initiation of each national lockdown and HCU.

The effect of the initiation of the third national lockdown was not estimated since the weeks prior coincided with Christmas, where utilisation is expected to be reduced. The prepandemic weeks (prior to 9 March 2020) were compared against each of the national lockdown periods to determine if there was a significant change in the rates of HCU associated with each of the national lockdowns using Poisson regression. A Poisson regression model, linear in time, was fit to the weekly rates of utilisation after the initiation of the first national lockdown until the end of the study to determine the overall change of utilisation throughout the pandemic. A direct comparison between utilisation observed in a calendar week in 2021 vs 2020 was conducted for calendar weeks 2–11, using RRs; it was assumed that the data in calendar weeks 2–11 in 2020 were unaffected by the pandemic and consequently act as a control. Additionally, the RRs between utilisation measures in the final 4 weeks of the study and the prepandemic period were estimated using Poisson regression to compare how utilisation differed from prepandemic levels by the end of the study.

Subgroup analyses were performed across LTC and IMD groups. To compare subgroups, we further provided the rates of utilisation per 1000 people by dividing by the total number of people assigned to the corresponding subgroup and multiplying by 1000. For example, when comparing the rates across number of LTCs (none, single or multiple), for the people without any LTCs, the rate of weekly secondary care admissions is defined as the total number of admissions experienced by this subgroup within a given week divided by the total number of people within the subgroup, multiplied by 1000. The interactive effect between the each of the national lockdowns and subgroup HCU was estimated using log-linear regression.

A sensitivity analysis to adjust for deaths that occurred during the study was performed, where rates of utilisation were recalculated in July 2021 dividing by the total numbers of patients that were still alive (death-adjusted), and compared with the unadjusted rates. July 2021 was chosen since this was 1 month before the end of the study and therefore captured the majority of deaths that occurred throughout the study; hence, if no difference was observed between the death-adjusted and unadjusted rates, the unadjusted rates would pertain across all weeks. All analyses were performed in R V.4.0.0, using the packages 'tidyverse',[20] 'scales',[21] 'reshape2'[22] and 'cowplot'.[23]

### Patient and public involvement

Two public representatives provided input throughout the project. Both representatives gave their full support to the proposed project and are preparing a patient and public summary of the research for dissemination.

**Table 1** Long-term conditions (LTC) and social deprivation identified in the Greater Manchester population and the Manchester CCG subpopulation

| | GM (N=3 225 169) | % | Manchester CCG (N=693 749) | % |
|---|---|---|---|---|
| LTC group* | | | | |
| Cancer | 50 954 | 1.6 | 6307 | 0.9 |
| Cardiovascular | 561 195 | 17.4 | 72 912 | 10.5 |
| Endocrine | 393 274 | 12.2 | 64 557 | 9.3 |
| Gastrointestinal | 501 060 | 15.5 | 81 957 | 11.8 |
| Musculoskeletal or skin | 284 103 | 8.8 | 52 593 | 7.6 |
| Neurological | 46 672 | 1.4 | 7340 | 1.1 |
| Psychiatric | 854 454 | 26.5 | 143 707 | 20.7 |
| Renal or urological | 130 436 | 4.0 | 16 617 | 2.4 |
| Respiratory | 550 648 | 17.1 | 93 174 | 13.4 |
| Sensory Impairment or Learning Disability | 314 264 | 9.7 | 45 909 | 6.6 |
| Substance abuse | 115 532 | 3.6 | 24 068 | 3.5 |
| Number of LTCs | | | | |
| None | 1 530 501 | 47.5 | 399 618 | 57.6 |
| Single | 631 648 | 19.6 | 127 428 | 18.4 |
| Multiple | 1 063 020 | 33.0 | 166 703 | 24.0 |
| IMD group | | | | |
| 1–2 (most deprived) | 1 335 061 | 41.4 | 405 862 | 58.5 |
| 3–4 (highly deprived) | 675 296 | 20.9 | 185 118 | 26.7 |
| 5–6 (moderately deprived) | 424 139 | 13.2 | 67 192 | 9.7 |
| 7–10 (least deprived) | 788 553 | 24.4 | 35 087 | 5.1 |
| Missing | 2120 | 0.1 | 490 | 0.1 |

*These data represent the overall prevalence of each LTC in the population. Each individual can be represented in more than one LTC row.
CCG, clinical commissioning group; GM, Greater Manchester; IMD, Index of Multiple Deprivation.

## RESULTS

The total population captured within the GMCR includes 3 225 169 patients, of whom 693 749 were registered with a MCCG. The mean age of the population was 38.2 years old (SD 22.8), with 49.0% of the population registered as female. The majority of the population (64.5%) had a registered ethnicity as white. Asian or British Asian patients represented 11.5% of the population and black or black British patients represented 4.0%. Ethnicity data were not available for 12.0% of patients. The prevalence of LTCs is shown in table 1. The most common LTCs observed were psychiatric (GMCR: 26.5%, MCCG: 20.7%), cardiovascular (GMCR: 17.4%, MCCG: 10.5%), respiratory (GMCR: 17.1%, MCCG: 13.4%) and gastrointestinal (GMCR: 15.5%, MCCG: 11.8%). Levels of deprivation were high, with 41.4% of the GM population and the majority of those registered within MCCG (58.5%) residing in areas that are in the most deprived quintile (decile of 1 or 2; table 1).

### Overall primary HCU

There was a rapid decrease in all weekly primary HCU measures starting just prior to the first national lockdown

(figure 1). The largest drops in activity associated with the initiation of the first national lockdown were for recording of healthcare information (% drop (95% CI): BP: 82.4% (82.0% to 82.9%); BMI: 79.5% (78.8% to 80.1%); cholesterol 84.9% (84.2% to 85.5%); HbA1c 84.0% (83.3% to 84.6%); smoking status 62.2% (61.3% to 63.1%). There was still a significant drop in the new prescriptions but the change was proportionally smaller (24.7%; 95% CI 24.0% to 25.5%). These reductions were sustained throughout the first national lockdown (online supplemental table S3). The initiation of the second national lockdown was associated with an increase or no significant change in primary HCU (% increase (95% CI): new prescriptions: −0.3% (−1.3% to 0.6%); BP: 4.1% (2.2% to 6.1%); BMI: 2.4% (0.0% to 4.8%); cholesterol: 10.0% (7.3% to 12.8%); HbA1c: 6.5% (4.1% to 8.9%); smoking status: −0.2% (−2.1% to 1.8%)).

All primary care HCU increased with time from the initiation of the first national lockdown to the end of the study (annual RR (95% CI): new prescriptions: 1.189 (1.186 to 1.191; p<0.001); BP: 2.113 (2.104 to 2.122; p<0.001); BMI: 2.097 (2.086 to 2.108; p<0.001); cholesterol: 2.221 (2.209

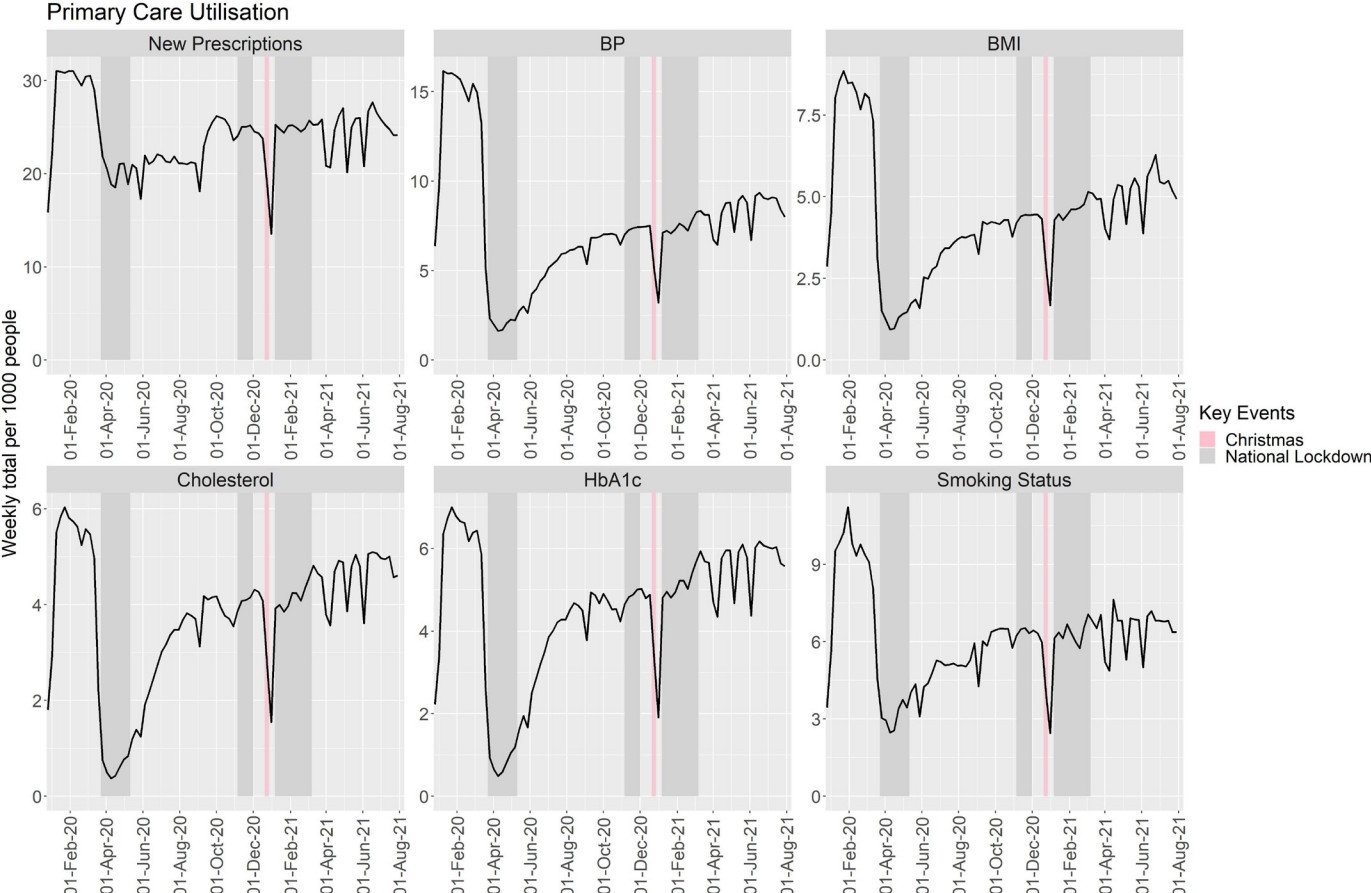

**Figure 1** Weekly primary care utilisation per 1000 people of the Greater Manchester population between January 2020 and August 2021. The first week covered 30 December 2019 to 5 January 2020, hence utilisation was expected to be considerably lower between this and the following week due to the UK bank holiday and seasonal effects expected for this calendar week. BMI, body mass index; BP, blood pressure; HbA1c, Haemoglobin A1c.

to 2.234; p<0.001); HbA1c: 2.165 (2.153 to 2.176; p<0.001); smoking status: 1.538 (1.531 to 1.544; p<0.001)). Despite this, by the end of the study, all measures were still recorded less often than in the prepandemic period (RR (95% CI; p value); new prescriptions: 0.828 (0.825 to 0.832; p<0.001); BP: 0.583 (0.580 to 0.587; p<0.001); BMI: 0.669 (0.664 to 0.675; p<0.001); cholesterol: 0.896 (0.888 to 0.905; p<0.001); HbA1c: 0.935 (0.927 to 0.943; p<0.001); smoking status: 0.711 (0.706 to 0.717; p<0.001)).

All primary care measures were lower across calendar weeks 2–11 when comparing 2021 data with 2020 data (p<0.001; online supplemental table S4). The measuring of BP and BMI remained consistently lower throughout these weeks by an average of 50.0% (95% CI 49.8% to 50.2%; p<0.001) and 42.5% (95% CI 42.1% to 42.8%; p<0.001), respectively. Even though the rates of cholesterol and HbA1c measurements taken in calendar week 11 were similar in 2021 (pandemic) and 2020 (prepandemic): 0.937 (95% CI 0.916 to 0.958) and 0.973 (95% CI 0.953 to 0.992), respectively, they were still significantly lower in 2021.

### Primary HCU by multi-morbidity and deprivation

Multi-morbid patients and patients with only a single LTC had consistently higher levels of primary HCU than patients with no LTCs throughout the study period (online supplemental table S5; figure 2). The ratio of weekly HCU rates per 1000 people between multi-morbid patients and those with no LTCs significantly increased for new prescriptions (RR 1.281; 95% CI 1.169 to 1.404; p<0.001), BP (RR 1.187; 95% CI 1.007 to 1.400; p=0.042) and smoking status (RR 1.356; 95% CI 1.126 to 1.632; p=0.002) during the first national lockdown, and decreased for BMI (RR 0.736; 95% CI 0.613 to 0.885; p=0.001) and smoking status (RR 0.803; 95% CI 0.675 to 0.956; p=0.014) in the third national lockdown. No significant changes in HCU between multi-morbid patients and those with no LTCs were observed for HbA1c or cholesterol or during the second national lockdown for all primary care HCU (online supplemental table S6 and figure S1).

### Primary HCU by deprivation

People who were from less deprived areas had lower rates of new prescriptions compared with those from the most deprived areas (IMD 1–2), (RR (95% CI); 3–4 vs 1–2: 0.915 (0.874 to 0.959); 5–6 vs 1–2: 0.920 (0.878 to 0.964); 7–10 vs 1–2: 0.875 (0.835 to 0.917); figure 2, online supplemental table S7). Similarly, smoking status had a lower rate of measurement in patients from the least deprived

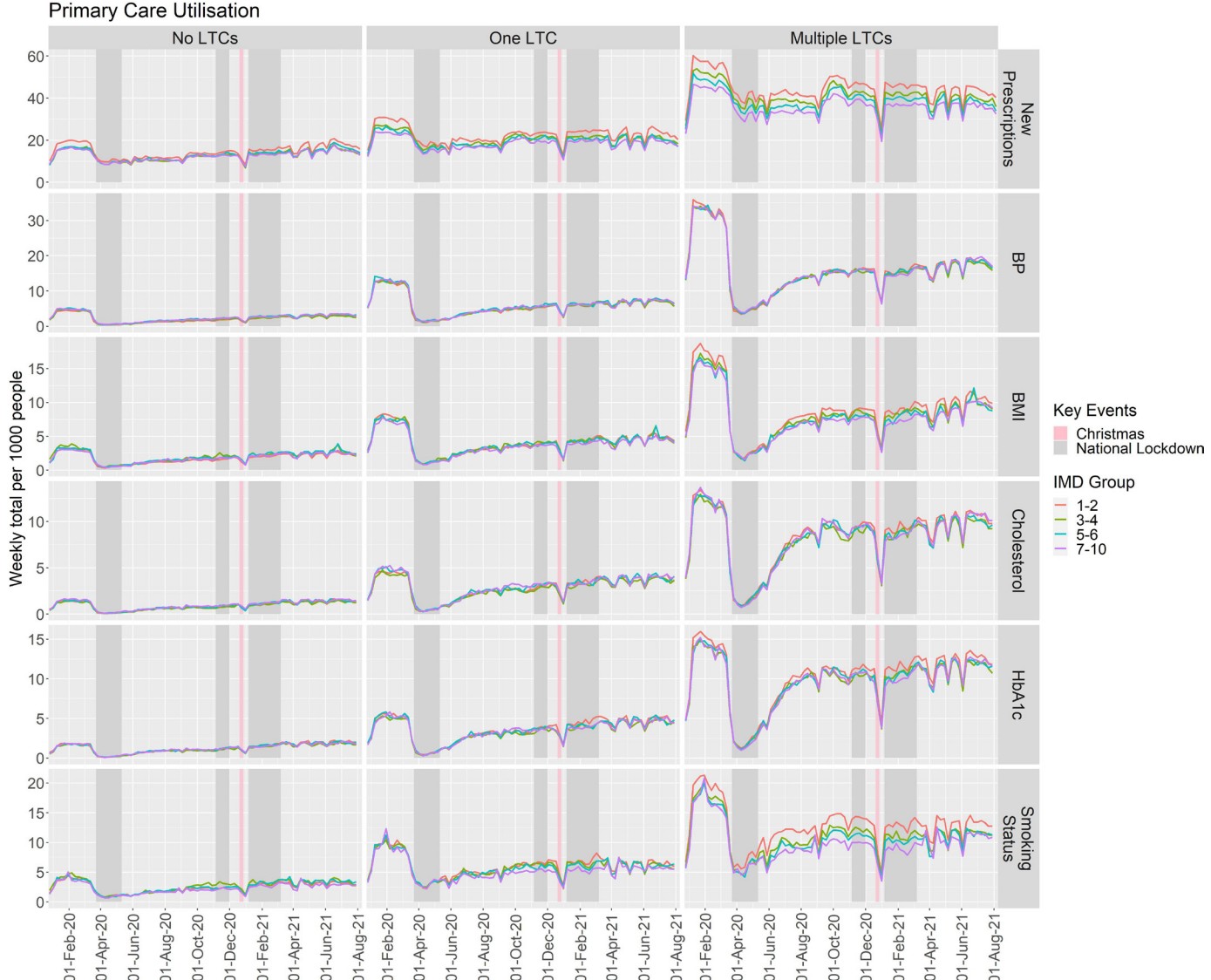

**Figure 2** Rates of primary care measures recorded per 1000 people per week, identified according to number of long-term conditions and deprivation group, between January 2020 and August 2021. BMI, body mass index; BP, blood pressure; HbA1c, Haemoglobin A1c; IMD, Index of Multiple Deprivation; LTC, long-term condition.

areas compared with patients from the most deprived areas (RR 0.885; 95% CI 0.801 to 0.877). No other differences were observed with regards to deprivation across primary HCU. The group from the least deprived areas experienced an additional reduction in smoking status measurement during the third national lockdown (RR 0.836; 95% CI 0.704 to 0.994; p=0.042) but no other interactions between deprivation and national lockdowns were evident (online supplemental table S8).

### Interaction between multi-morbidity and deprivation for primary HCU

Differences in HCU by deprivation were overall larger within multi-morbid patients (online supplemental table S9; figure 2). Differences in HCU between deprivation groups were not attributable to only one LTC group (online supplemental figure S2). In multi-morbid patients, there were no significant changes in the ratio of

weekly HCU per 1000 people between the group from the least deprived areas and the groups from other deprivation areas, across all primary HCU measures, during the first national lockdown compared with prepandemic weeks (online supplemental table S10 and figure S3).

### Overall secondary HCU

There has been large variation in planned and unplanned secondary HCU over the course of the COVID-19 pandemic (online supplemental figure S4). There was a 47.4% (95% CI 42.9% to 51.5%, p<0.001) reduction in planned and 35.3% (95% CI 28.3% to 41.6%; p<0.001) reduction in unplanned weekly admission rates per 1000 people associated with the initiation of the first national lockdown (admission rate prepandemic vs initiation of first national lockdown; planned: 2.51 vs 1.32; unplanned: 1.36 vs 0.88). The initiation of the second national lockdown was also associated with a significant reduction in

secondary HCU; planned weekly admission rates per 1000 people reduced by 20.4% (95% CI 14.4% to 25.9%; p<0.001) and unplanned reduced by 15.6% (95% CI 7.3% to 23.1%; p<0.001). The reductions were sustained throughout these lockdowns; only unplanned admissions in the third national lockdown were not significantly lower than prepandemic rates (online supplemental table S3). The patterns observed in secondary admissions were consistent across all three main contributing secondary care providers (online supplemental figure S5).

Both planned and unplanned weekly admissions were on average lower from the beginning of the first national lockdown up until the end of the study period, compared with the prepandemic admissions (planned: RR 0.850, 95% CI 0.837 to 0.864, p<0.001; unplanned: RR 0.976, 95% CI 0.957 to 0.996, p=0.016). However, the admissions increased throughout the period (planned: p<0.001; unplanned: p<0.001) and when comparing the final 4 weeks of the study period with the prepandemic period, planned admission rates were not significantly different (RR 1.105; 95% CI 0.987 to 1.044; p=0.290) and unplanned were higher (RR 1.104, 95% CI 1.067 to 1.143; p<0.001). The direct comparison between calendar weeks 2–11 in 2021 vs 2020 indicated that planned admissions were lower on average by 11.3% (95% CI 9.4% to 13.2%; p<0.001) but there was no difference in unplanned admissions (RR 1.012; 95% CI 0.985 to 1.040; p=0.376). A week-by-week comparison is detailed in online supplemental table S3.

### Secondary HCU by multi-morbidity

Morbidity was associated with an increased rate of planned admissions throughout the study period: single vs no LTCs RR 1.904 (95% CI 1.717 to 2.111; p<0.001); multiple vs no LTCs RR 9.584 (95% CI 8.644 to 10.627; p<0.001); multiple vs single LTC RR 5.033 (95% CI 4.540 to 5.581; p<0.001). This was also the case for unplanned admissions: single vs no LTCs RR 1.188 (95% CI 1.112 to 1.270; p<0.001); multiple vs no LTCs RR 3.636 (95% CI 3.401 to 3.887; p<0.001); multiple vs single LTC RR 3.059 (95% CI 2.862 to 3.271; p<0.001) (figure 3).

While the ratio of weekly unplanned admissions per 1000 people between patients that were multi-morbid versus those without any LTCs was consistent throughout the majority of the pandemic, there was a significant increase during the first lockdown compared with prepandemic (RR 1.253, 95% CI 1.068 to 1.469; p=0.006) but no significant change was observed between those with a single LTC and those without any LTCs (RR 0.987, 95% CI 0.842 to 1.157; p=0.865, online supplemental figure S6). The ratio of planned admission rates per 1000 people in morbidity groups increased during the first national lockdown compared with that observed prepandemic: multiple vs no LTCs increased by a factor of 2.402 (95% CI 2.047 to 2.818; p<0.001) and single vs no LTCs increased by a factor of 1.413 (95% CI 1.205 to 1.658; p<0.001) (online supplemental table S5 and figure S6), however, this was not sustained throughout the pandemic.

The average of the ratios of admission rates between multi-morbid patients versus patients without LTCs from the start of the first national lockdown until the end of the study period versus prepandemic was 1.176 (95% CI 0.871 to 1.588; p=0.289) for planned admission rates and 1.097 (95% CI 0.900 to 1.338; p=0.357) for unplanned admission rates.

### Secondary care HCU for specific LTC groups

There were noticeable differences for both planned and unplanned admission rates within each LTC group over the study period (figure 4). Planned admission rates were highest for patients with a renal or urological LTC. Unplanned admission rates were highest in patients with cancer or renal or urological LTCs. Planned and unplanned admission rates were lowest overall for patients without any LTC. For patients with cancer, the drop in the number of planned admissions at the initiation of the first national lockdown was sustained throughout the remainder of the study period, with an average reduction of 28.9% (95% CI 25.1% to 32.5%; p<0.001) compared with prepandemic levels and unplanned admission rates decreased by 11.5% (95% CI 2.4% to 19.6%; p=0.014). Planned admission rates for people identified as having an endocrine, musculoskeletal or skin, neurological, psychiatric or respiratory LTC returned to prepandemic levels by the end of the study period. However, planned admission rates for people identified as having cancer, cardiovascular, gastrointestinal, renal or urological and sensory impairment or learning disability remained lower than in the prepandemic period. Conversely, planned admission rates for people identified with a substance abuse LTC were higher by the end of the study period compared with the prepandemic period (RR 1.196; 95% CI 1.074 to 1.329; p=0.001; online supplemental table S11). Unplanned admissions rates were lower only for those that were identified with a renal or urological LTC. Patient groups with a gastrointestinal, musculoskeletal or skin, psychiatric or substance abuse LTC had higher rates of unplanned admissions at the end of the study compared with prepandemic levels. The remaining LTC groups had no significant change in unplanned admissions (online supplemental table S11).

### Secondary HCU by deprivation

People from the most deprived areas (IMD of 1 or 2) had the highest rates for both planned (RR (95% CI) 3–4 vs 1–2: 0.753 (0.694 to 0.818; p<0.001); 5–6 vs 1–2: 0.787 (0.724 to 0.854; p<0.001); 7–10 vs 1–2: 0.812 (0.748 to 0.882; p<0.001)) and unplanned admissions (RR (95% CI) 3–4 vs 1–2: 0.686 (0.642 to 0.732; p<0.001); 5–6 vs 1–2: 0.670 (0.628 to 0.715; p<0.001); 7–10 vs 1–2: 0.683 (0.640 to 0.729; p<0.001)) throughout the study period. For multi-morbid patients, being from a highly deprived area was associated with an increased rate in both planned and unplanned admissions compared to all other deprivation areas (online supplemental table S3c

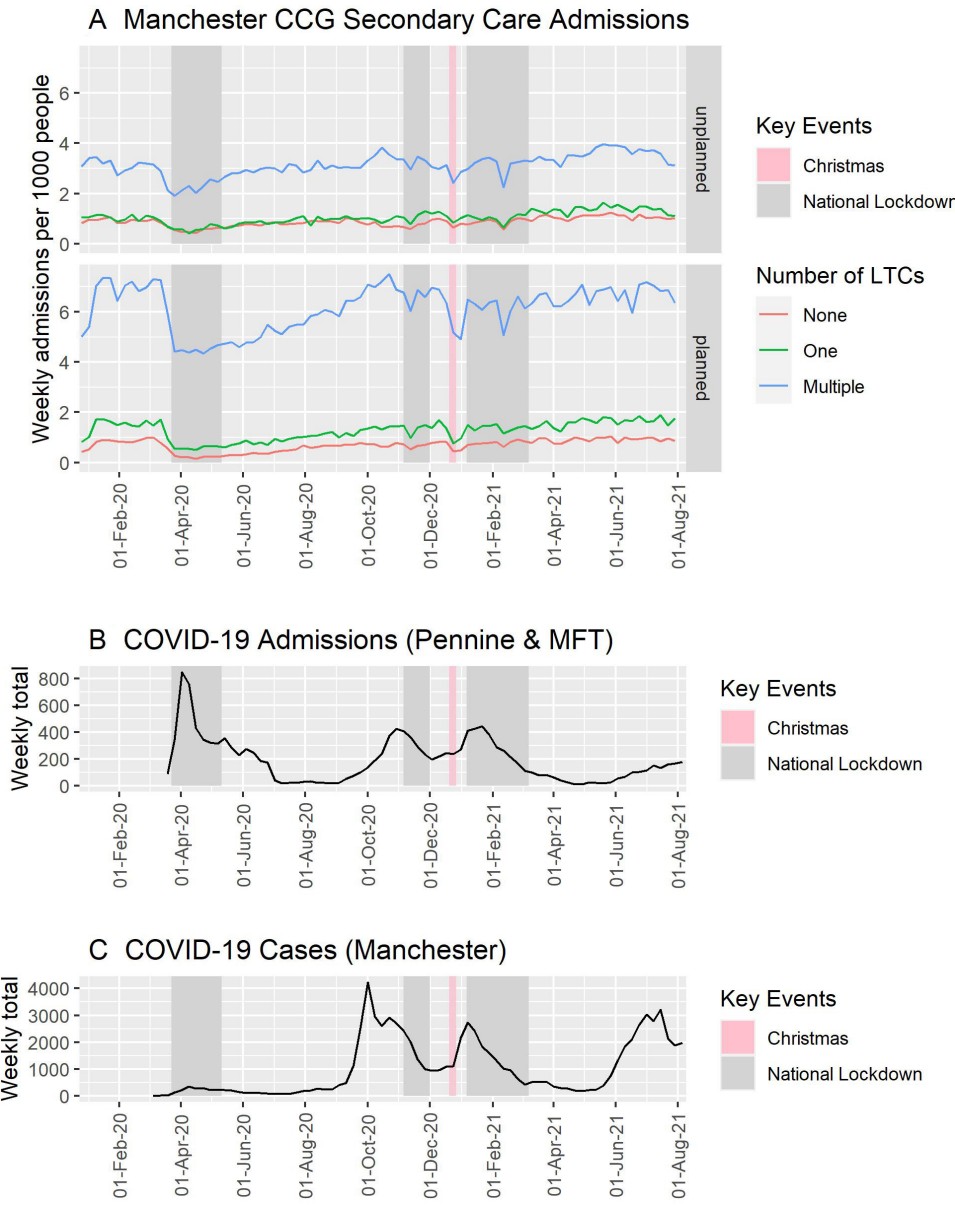

**Figure 3** (A) Weekly rates of planned and unplanned admissions per 1000 people that were identified as having zero (none), one (single) or multiple LTCs, within the Manchester CCG subpopulation between January 2020 and August 2021. (B) Government reported COVID-19 admissions in Manchester University NHS Foundation Trust, Pennine Acute Hospitals NHS Foundation Trust and Pennine Care NHS Foundation Trust (extracted 8 September 2021) and (C) Government reported cases in Manchester (extracted 8 September 2021). CCG, clinical commissioning group; LTC, long-term condition; MFT, Manchester University Hospital Foundation Trust; NHS, National Health Service; Pennine, Pennine Acute Hospitals NHS Trust & Pennine Care NHS Foundation Trust.

and figure S7). The ratios of rates between deprivation area groups within multi-morbid patients were not significantly different during the first national lockdown compared to prepandemic levels (online supplemental table S10 and figure S8).

## DISCUSSION
### Principal findings and interpretation
We have assessed primary HCU for over 3 million patients across GM and secondary HCU for a subgroup of almost 700 000 patients within MCCG. Major

changes in HCU occurred during the COVID-19 pandemic. There was a large reduction in both primary and secondary HCU at the beginning of the first national lockdown. While there was a relatively consistent increase in primary care HCU from the first national lockdown, primary HCU remained lower at the end of the study compared with prepandemic. Overall, both planned and unplanned secondary admissions had recovered to prepandemic levels by the end of the study period but this recovery was not observed across all LTC subgroups. Changes in the

MCCG Secondary Care Admissions

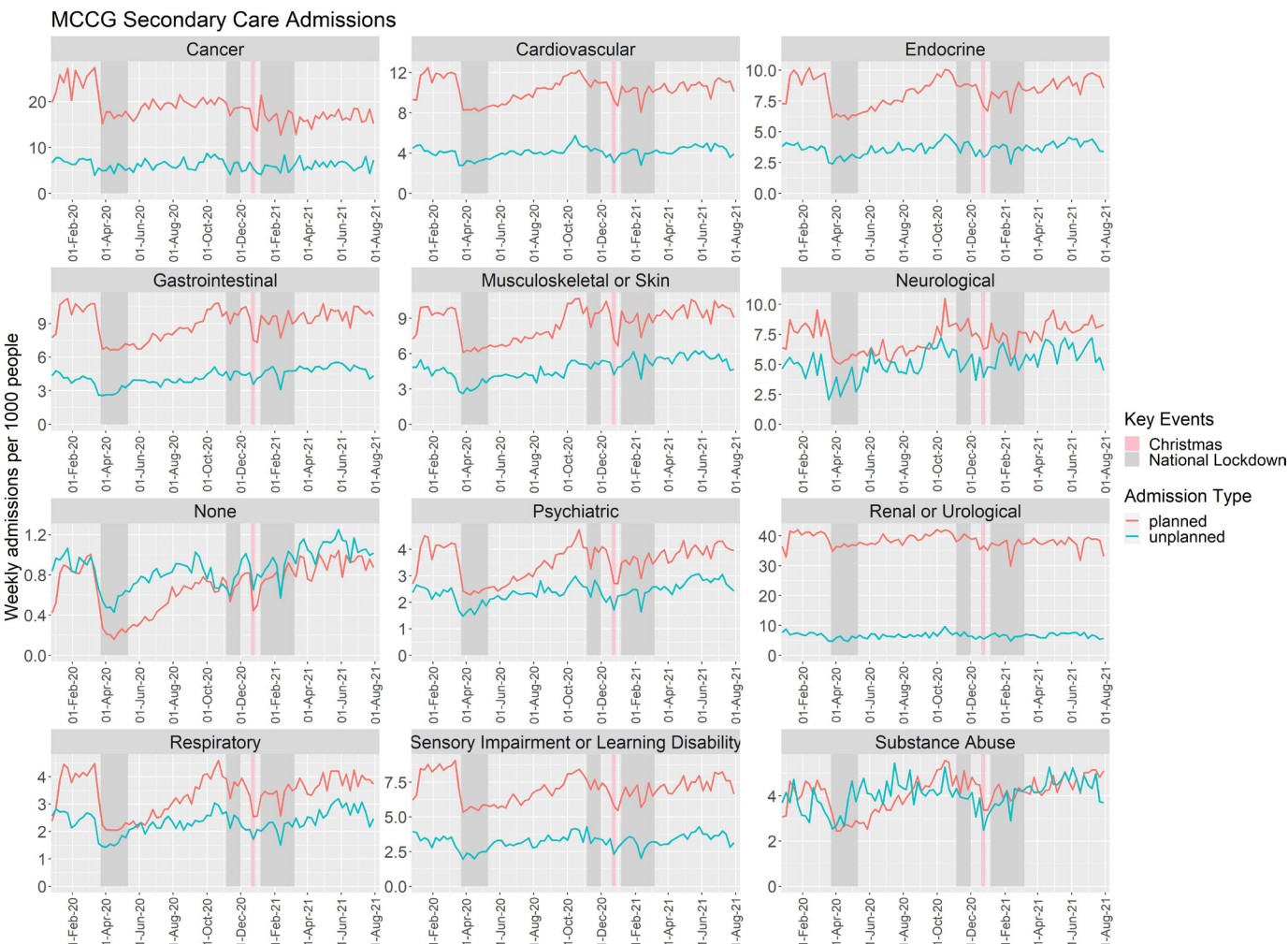

**Figure 4** Weekly rates of planned and unplanned admissions identified in patients with each of the long-term conditions within the Manchester clinical commissioning group (MCCG) subpopulation, between January 2020 and August 2021.

ratio of HCU between multi-morbid patients and those without LTCs occurred during national lockdowns but were inconsistent across primary HCU measures.

Although some healthcare information measures can be completed remotely (eg, smoking status, BP and BMI), primary HCU measures that require in-person contact with a healthcare professional (eg, HbA1c and cholesterol) demonstrated similar patterns in HCU. The initial larger fall in healthcare information recording compared with incident prescribing in primary care may reflect a shift in focus away from secondary prevention during the first wave of the pandemic. It is also possible that coding practices may have changed with the switch from face-to-face to remote consultations and that this switch has also impacted on opportunistic BP/BMI/smoking status checks. Although these HCU measures have not returned to prepandemic levels, they have consistently increased since the first lockdown and this has occurred even though quality outcome framework targets and local enhanced services were largely suspended.

Despite a peak in COVID-19 admissions within the first national lockdown, secondary admissions fell by a larger volume. Reductions in secondary care admissions

associated with the first lockdown have been reported across the UK.[11 24 25] Largely, these are reflective of cancellations of elective activity or delaying non-urgent care, to ensure capacity for patients with severe COVID-19 infection and to increase critical care capacity.[7] The observed deficit may not correspond entirely to an unmet need for patients with non-COVID-19 related healthcare needs as there is some evidence that changes in behaviour according to sanitisation campaigns, social distancing and government restrictions may have resulted in fewer infections,[26] and injuries.[27] Additionally, emergency department attendances which are related to unplanned admissions (but were not directly assessed in this study) have been observed to have reduced.[14] It is also possible that the increased utilisation of remote management for secondary care patients has contributed to clinically appropriate reductions in admissions.

There has been no noticeable recovery in HCU for patients with cancer and for a number of other LTCs, recovery to prepandemic HCU levels has not occurred. In contrast, HCU of patients identified with substance abuse and/or a psychiatric condition exceeded prepandemic levels between the first and second national lockdowns,

likely reaffirming the significant impact of the pandemic on mental health and psychiatric services.[28]

## Implications for clinicians and policy-makers

It is inevitable that the changes in HCU observed in this study will have had an impact on both patients and healthcare providers above and beyond the direct impact of COVID-19. For patients with cancer, services had to adapt to mitigate the increased risk of death from COVID-19.[29] The initial reduction in the number of planned admissions was sustained throughout the study period and is likely to reflect changes in services but may also be due to patients with cancer being reluctant to seek healthcare. Delays in care for patients with cancer are known to impact prognosis,[30] and the pandemic has been found to have contributed to excess deaths in patients with cancer.[24] A proactive approach to encourage patients to attend screening and routine appointments will be needed to minimise the impact of the pandemic on patients with cancer and other emerging health inequalities.[31 32] Understanding the implications of reductions in the selected primary care HCU measures, particularly the decrease in assessing and recording healthcare information will require further long-term studies.

## Strengths and limitations

The strengths of this study include the complete coverage of a large geographical area for the primary care analyses and the inclusion of both primary and secondary HCU data. This is the first study to evaluate HCU across the full spectrum of LTC subpopulations and stratify according to multi-morbidity and deprivation. Data prior to 2020 were not available and consequently comparisons made (prepandemic vs postpandemic) are reliant on the data between January and March 2020 being representative of prepandemic utilisation. Consequently, the comparison of prepandemic HCU to the end of the study period may have been influenced by seasonal variations in HCU. The secondary care analysis was only possible on a subset of the GM population due to delays in data from some GM secondary care providers. A most common provider method was used to assign patients for the secondary care analysis. A limitation of this approach is that patients may travel for secondary care to different hospitals outside of their assigned provider. While this limitation may impact on the rates of utilisation per 1000 people, it is highly unlikely to have caused variation in the rate over time.

The study population represents a largely deprived area placed under strict restrictions during the pandemic. While this information is valuable, the findings may not be generalisable to other settings in the UK or internationally. Although the measures of HCU that have been selected are relevant and reliable, they do not provide a complete picture of either primary or secondary HCU. There is no single effective measure to summarise HCU in primary care as there are many aspects that reflect HCU in this setting.[33] It remains possible that the shift towards increased remote consultations may have resulted in changes to primary care delivery that were not possible to accurately capture using our measures of primary HCU. Additionally, the cause of admission was not available for secondary HCU, hence we were unable to determine LTC-specific admissions. While the current scaling and subpopulations do not take into consideration any deaths or new diagnoses that occurred after 1 January 2020, a sensitivity analysis accounting for deaths resulted in very small increases to rates (online supplemental figures S9, S10 and table S12) and scaled utilisation remained lower than prepandemic levels.

## CONCLUSIONS

We have assessed the changes in HCU in primary and secondary care associated with the COVID-19 pandemic and UK national lockdowns for patients with LTCs across a large urban region. There was a significant reduction in both primary and secondary HCU associated with the first national lockdown. Subsequent national lockdowns were associated with reductions in secondary care but not in primary care. While some measures of HCU had returned to prepandemic levels by the end of the study, many had not. Proportionally, secondary care HCU increased in multi-morbid patients compared with those without LTCs during the first and second national lockdowns. Although changes to HCU during the pandemic have been similar overall, different patterns have been seen in specific LTC groups such as people with cancer. Over the course of the pandemic deprivation was associated with higher rates of HCU in multi-morbid patients but no significant differences were observed in the ratio of utilisation between those residing in the most and least deprived areas for the majority of HCU measures during national lockdowns.

**Author affiliations**
¹Division of Informatics, Imaging and Data Science, Faculty of Biology Medicine and Health, The University of Manchester, Manchester, UK
²National Institute for Health Research Applied Research Collaboration Greater Manchester, The University of Manchester Faculty of Biology Medicine and Health, Manchester, UK
³National Institute for Health Research Greater Manchester Patient Safety Translational Research Centre, The University of Manchester Faculty of Biology Medicine and Health, Manchester, UK
⁴Langworthy Medical Practice, Salford, UK
⁵National Institute for Health Research Manchester Biomedical Research Centre, Faculty of Biology, Medicine and Health, The University of Manchester, Manchester, UK
⁶Division of Cardiovascular Sciences, The University of Manchester, Manchester, UK

**Acknowledgements** We would like to thank Eric Lowndes and Sumaira Naseem for providing the public representation throughout our research. The authors recognise the Greater Manchester Care Record (a partnership of Greater Manchester Health and Social Care Partnership, Health Innovation Manchester and Graphnet Health, on behalf of Greater Manchester localities) in the provision of data required to undertake this work. Using patient data is vital to improve health and care for everyone. There is huge potential to make better use of information from people's patient records, to understand more about disease, develop new treatments, monitor safety and plan NHS services. Patient data should be kept safe and secure, to protect everyone's privacy, and it is important that there are

safeguards to make sure that it is stored and used responsibly. Everyone should be able to find out about how patient data is used.

**Contributors** SWG conceived the study. NP and RW led on the acquisition of data for the study. SWG, NP, RW, OT, MS and CS-P were involved in the design of the study and interpretation of the data. CS-P led the analysis of the data, supported by MS. CS-P and SG prepared the manuscript. All authors revised the manuscript and approved the final version. SWG is the guarantor for this study.

**Funding** The time of RW and NP was partially funded by the National Institute for Health Research (NIHR) Greater Manchester Patient Safety Translational Research Centre (PSTRC-2016-003) and the NIHR Applied Research Collaboration Greater Manchester (NIHR200174). The time of NP was partially funded by the NIHR Manchester Biomedical Research Centre (IS-BRC-1215-20007). The time of CS-P was fully funded by the NIHR Applied Research Collaboration Greater Manchester (NIHR200174).

**Disclaimer** The views expressed are those of the author(s) and not necessarily those of the NIHR or the Department of Health and Social Care.

**Competing interests** None declared.

**Patient and public involvement** Patients and/or the public were involved in the design, or conduct, or reporting, or dissemination plans of this research. Refer to the Methods section for further details.

**Patient consent for publication** Not applicable.

**Ethics approval** The study was approved by both the GMCR Expert Review Group and Research Governance Group. All data made available to the analysts were deidentified and aggregated and therefore did not require specific ethical approval.

**Provenance and peer review** Not commissioned; externally peer reviewed.

**Data availability statement** No data are available. Data and code were hosted in a secure environment and are not freely available to share.

**ORCID iDs**
Camilla Sammut-Powell http://orcid.org/0000-0002-5855-1492
Matthew Sperrin http://orcid.org/0000-0002-5351-9960

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
