## [Reviewer comments · BMJ Open]

ARTICLE DETAILS

TITLE (PROVISIONAL)	Healthcare utilisation in patients with long-term conditions during the COVID-19 pandemic: a population-based observational study of all patients across Greater Manchester, UK
AUTHORS	Sammut-Powell, Camilla; Williams, Richard; Sperrin, Matthew; Thomas, Owain; Peek, N; Grant, Stuart

VERSION 1 – REVIEW

REVIEWER	Beane, Thomas Imperial College London, Department of Primary Care and Public Health
REVIEW RETURNED	06-Oct-2022

GENERAL COMMENTS	This paper reports on a well-conducted study of healthcare utilisation from the start of the pandemic. Although other studies have reported changes in health service utilisation during the pandemic, this study uniquely describes the changes across both primary and secondary care within a single regional population and across a longer time period than many other studies. As a result, the findings would be of interest to a wide audience. However, this also means it could easily have been two papers, and there is a lot to fit into the results. I do feel the paper would benefit from cutting down on some of the results section, particularly for the subgroup analyses. Overall the paper is well written and clearly reported. I have more specific comments below Methods: • Was there a reason appointment data could not be used for primary care? If not available, this should be clarified, as it would seem the obvious first choice metric for utilisation.• P6, I'm unclear on the measure of incident prescriptions – is this 'new' in the sense of a drug being prescribed for the first time ever or within some timeframe? For example, if an antibiotic was prescribed a month ago and then prescribed again, does this count as new? Results • Table 1: it would be useful to present the age and sex distribution (and ethnicity if available) of the population to help judge how generalisable the results are to other populations. Were these factors included in the regression models?• P7, lines 51-54: these estimates are hard to interpret as so close to 1 when rounded – the first in particular doesn't seem to support a moderate increase. Could they be instead written as % increases in the RR, or aggregated over a longer time period, eg month?
---

	 • P9, line 33 – an important distinction for deprivation is that it is not the ‘people that were less deprived’, at least when using IMD, but people living in more deprived areas – there is often considerable variation at an individual level within LSOAs. There are a few places where this should be amended. Discussion:  • P12. Lines 7-16 – while blood tests indicate an appointment was needed, and a reduction does indicate a fall in blood test appointments, falls in smoking/BP/BMI codes + prescribing does not necessarily indicate a fall in GP consultations. Instead it could reflect changes in how clinicians are coding information remotely versus F2F, rather than a fall in consultations/utilisation. • Following on from the above, it could reflect a shift in focus from secondary prevention, but I suspect this is less of a shift, but more the case that fewer opportunistic BP/BMI checks can be taken remotely compared to when seen F2F. • Limitations: the secondary care denominators used a most common provider method to assigning the population. Particularly within urban areas, people may be more willing to travel more to different hospitals and so there can be considerable overlap in the populations assigned to different Trusts (see https://www.ncbi.nlm.nih.gov/pmc/articles/PMC8753015/). I agree with the approach you used, but it is a limitation that should be acknowledged, as hospitals do not have distinct geographic boundaries. • Limitations: I note that LTCs were calculated as of 1st January 2020 – this will mean that patients newly developing LTCs will not be counted – and potentially may dilute effect sizes for this group?
--	--

REVIEWER	Alamrawy, Roa Mamoura Psychiatric Hospital, General Secretariat of Mental Health and Addiction Treatment
REVIEW RETURNED	12-Nov-2022

GENERAL COMMENTS	The article is really well written. Huge efforts were reflected in the statistical part and hence the discussion. I see that including some implications of this research work in the abstract, makes it more appealing to be read and spots the light on the relevance of those analyses to practice and future directions.
--

VERSION 1 – AUTHOR RESPONSE

REVIEWER 1 COMMENTS

Comment

This paper reports on a well-conducted study of healthcare utilisation from the start of the pandemic. Although other studies have reported changes in health service utilisation during the pandemic, this study uniquely describes the changes across both primary and secondary care within a single regional population and across a longer time period than many other studies. As a result, the findings would be of interest to a wide audience. However, this also means it could easily have been two papers, and there is a lot to fit into the results. I do feel the paper would benefit from cutting down on some of the results section, particularly for the subgroup analyses. Overall the paper is well written and clearly reported. I have more specific comments below

Response

We thank the reviewer for their comment. While we agree that there is a lot to fit in the results we think that separating out any more of the results into the supplementary material would detract from the thorough nature of the manuscript. Two manuscripts could have been considered but we felt that a complete overview of the topic in one manuscript would be of significant interest to readers.

Comment

Was there a reason appointment data could not be used for primary care? If not available, this should be clarified, as it would seem the obvious first choice metric for utilisation.

Response

Appointment data was not available in the GMCR which is why the surrogate markers of health care utilisation were used in the primary care analyses. We have added the information on lack of appointment data to the methods section (page 5, paragraph 4):

'For primary HCU, appointment data were not available within the GMCR. Surrogate markers of HCU were therefore evaluated consisting of first prescriptions and recording of healthcare information in the GP record.'

Comment

P6, I'm unclear on the measure of incident prescriptions – is this 'new' in the sense of a drug being prescribed for the first time ever or within some timeframe? For example, if an antibiotic was prescribed a month ago and then prescribed again, does this count as new?

Response

Thank you for raising this important point. We agree that that the definition could be more clearly described and have amended this in the text by adding the term 'non-repeat prescription' (page 5 paragraph 4). Regarding this specific example the second prescription would count as a new prescription.

'First prescriptions were identified by the issuing of any new prescription (non-repeat prescription) for an individual patient by a primary care healthcare professional and are referred to as incident prescriptions throughout the manuscript.'

Comment

Table 1: it would be useful to present the age and sex distribution (and ethnicity if available) of the population to help judge how generalisable the results are to other populations. Were these factors included in the regression models?

Response

We thank the reviewer for their comment. These were not included in the regression models as they would have been constant over time in the way that the analysis was performed due to only having weekly aggregate data as opposed to individual participant data. We have now added age, sex and ethnicity data to the results (page 6, paragraph 6) to allow for an understanding of generalisability of the population:

'The mean age of the population was 38.2 (SD 22.8), with 49.0% of the population registered as female. The majority of the population (64.5%) had a registered ethnicity as white. Asian or British Asian patients represented 11.5% of the population and Black or Black British patients represented 4.0%. Ethnicity data were not available for 12.0% of patients.'

Comment

P7, lines 51-54: these estimates are hard to interpret as so close to 1 when rounded – the first in particular doesn't seem to support a moderate increase. Could they be instead written as % increases in the RR, or aggregated over a longer time period, eg month?

Response

We thank the reviewer for this comment and have changed the time period to be over 1 year instead. We believe this provides a clearer indication of the increase and remains consistent in the presentation of the RRs with the rest of the manuscript (page 8, paragraph 1):

'All primary care HCU increased with time from the initiation of the first national lockdown to the end of the study (annual RR (95%CI): new prescriptions: 1.189 (1.186 - 1.191; p<0.001); BP: 2.113 (2.104 – 2.122;p<0.001); BMI: 2.097 (2.086 – 2.108;p<0.001); Cholesterol: 2.221 (2.209 – 2.234;p<0.001); HbA1c: 2.165 (2.153 – 2.176;p<0.001); Smoking status: 1.538 (1.531 – 1.544;p<0.001)).'

Comment

P9, line 33 – an important distinction for deprivation is that it is not the 'people that were less deprived', at least when using IMD, but people living in more deprived areas – there is often considerable variation at an individual level within LSOAs. There are a few places where this should be amended.

Response

We thank the reviewer for their insightful comment and have corrected this phrasing throughout the manuscript, e.g.

'People that were from less deprived areas had lower rates of new prescriptions compared to the most deprived areas (IMD 1-2), (RR (95%CI); 3-4 vs 1-2: 0.915 (0.874 – 0.959); 5-6 vs 1-2: 0.920 (0.878 – 0.964); 7-10 vs 1-2: 0.875 (0.835 – 0.917); Figure 2, Supplementary Table S7).'

Comment

P12. Lines 7-16 – while blood tests indicate an appointment was needed, and a reduction does indicate a fall in blood test appointments, falls in smoking/BP/BMI codes + prescribing does not necessarily indicate a fall in GP consultations. Instead it could reflect changes in how clinicians are coding information remotely versus F2F, rather than a fall in consultations/utilisation. Following on from the above, it could reflect a shift in focus from secondary prevention, but I suspect this is less of a shift, but more the case that fewer opportunistic BP/BMI checks can be taken remotely compared to when seen F2F.

Response

We thank the reviewer for this comment. While we think a change in coding practices is unlikely to have impacted on the results, it is a possibility and we have therefore added this to the second paragraph of the discussion. We have also commented on a reduction in opportunistic BP/BMI checks (page 11, paragraph 3):

'It is also possible that coding practices may have changed with the switch from face-to-face to remote consultations and that this switch has also impacted upon opportunistic BP/BMI/smoking status checks.'

Comment

Limitations: the secondary care denominators used a most common provider method to assigning the population. Particularly within urban areas, people may be more willing to travel more to different hospitals and so there can be considerable overlap in the populations assigned to different Trusts (see <https://www.ncbi.nlm.nih.gov/pmc/articles/PMC8753015/>). I agree with the approach you used, but it is a limitation that should be acknowledged, as hospitals do not have distinct geographic boundaries.

Response

We thank the reviewer for their comment. While we acknowledge that this is a limitation in precisely estimating the secondary HCU per 1000 people we feel that it is unlikely to have impacted

significantly upon the trend analyses. We have added this as limitation to the discussion (page 12, paragraph 3):

'A most common provider method was used to assign patients for the secondary care analysis. A limitation of this approach is that patients may travel for secondary care to different hospitals outside of their assigned provider. While this limitation may impact on the rates of utilisation per 1000 people, it is highly unlikely to have caused variation in the rate over time.'

Comment

Limitations: I note that LTCs were calculated as of 1st January 2020 – this will mean that patients newly developing LTCs will not be counted – and potentially may dilute effect sizes for this group?

Response

The reviewer is correct and we have already identified this as a limitation and undertaken a sensitivity analysis to accommodate it (page 12, paragraph 1).

'Whilst the current scaling and sub-populations do not take into consideration any deaths or new diagnoses that occurred after 1 Jan 2020, a sensitivity analysis accounting for deaths resulted in very small increases to rates (Supplementary Figures S9-S10; Supplementary Table S12) and scaled utilisation remained lower than pre-lockdown levels.'

REVIEWER 2 COMMENTS

Comment

The article is really well written. Huge efforts were reflected in the statistical part and hence the discussion. I see that including some implications of this research work in the abstract, makes it more appealing to be read and spots the light on the relevance of those analyses to practice and future directions.

Response

We thank the reviewer for their kind comments.

VERSION 2 – REVIEW

REVIEWER	Beaney, Thomas Imperial College London, Department of Primary Care and Public Health
REVIEW RETURNED	11-Feb-2023
GENERAL COMMENTS	All my comments have been addressed in the responses and in the manuscript, many thanks to the authors for this important work.